# Prevalence and Risk Factors for Self-Report Diabetes Mellitus: A Population-Based Study

**DOI:** 10.3390/ijerph17186497

**Published:** 2020-09-07

**Authors:** Isabela Silva Levindo de Siqueira, Rafael Alves Guimarães, Samira Nascimento Mamed, Thays Angélica de Pinho Santos, Suiany Dias Rocha, Valéria Pagotto, Karlla Antonieta Amorim Caetano, Claci Fátima Weirich Rosso

**Affiliations:** 1Faculty of Nursing, Federal University of Goiás, Goiânia 74605-080, Brazil; isabelalevindo@gmail.com (I.S.L.d.S.); thayssantos.ifg@gmail.com (T.A.d.P.S.); valeriapagotto@gmail.com (V.P.); karlla@ufg.br (K.A.A.C.); claci@ufg.br (C.F.W.R.); 2School of Social and Health Sciences, Pontifical Catholic University of Goiás, Goiânia 74175-120, Brazil; samiramamed31@gmail.com; 3Municipal Health Secretariat, Goiânia 74884-900, Brazil; 4Federal Institute of Goiás, Goiânia 74055-110, Brazil; 5Municipal Education Secretariat, Goiânia 74610-130, Brazil; suianydias@gmail.com

**Keywords:** diabetes mellitus, risk factors, epidemiology, Brazil

## Abstract

The aim of this study was to estimate the prevalence and risk factors for self-reported diabetes mellitus (DM) in the adult population of the Central-West region of Brazil. In 2013, a cross-sectional study using the data from the National Health Survey and comprising 7519 individuals aged ≥18 years from the Central-West region was conducted. Participants were interviewed at their homes about sociodemographic data and risk factors for DM. To verify the risk factors with DM, the Poisson regression model was used. The analyses were performed for the total sample and stratified according to sex. The prevalence of DM was 6.5% (95% confidence interval [95% CI], 5.7–7.3). The diagnosis of self-reported DM was 4.3% in men and 7.5% in women. In the global sample, it was found that age between 40–59 years and ≥60 years, previous smoking (former smoker), self-reported hypertension, self-reported dyslipidemia, overweight, and obesity were independently associated with self-reported DM. In men, risk factors were: Age ≥ 60 years, self-reported hypertension, self-reported dyslipidemia, and obesity. In women, risk factors were: Age 30–39 years, 40–59 years, and ≥60 years, previous smoking (former smoker), self-reported hypertension, self-reported dyslipidemia, overweight, and obesity. Conclusion: The prevalence of DM was 6.5%. DM was associated with advanced age; previous smoking (former smoker), hypertension, dyslipidemia, overweight, and obesity. Some differences in risk factors between men and women were noted.

## 1. Introduction

Diabetes mellitus (DM) is considered a major global public health problem. In 2014, according to the estimates of the World Health Organization (WHO), the prevalence of diabetes was 8.5% in adults, representing 422 million people diagnosed with this disease worldwide [1]. Considering its exponential growth, the proportion of people with DM aged 20 to 79 years increased to 425 million in 2017 (8.8%), with an estimated prevalence of 2045 at 9.9% [2]. In 2017, diabetes was estimated to have caused 5 million deaths and was responsible for 89 million disability-adjusted life years [2,3].

In Brazil, DM is also significantly prevalent and represents almost 5% of the disease burden in the country [4]. The significant economic impact of this disease extends to individuals, families, communities, healthcare systems, and countries as a whole [4,5]. According to the WHO data, the estimated prevalence of DM was 8.1% (7.4% in men and 8.8% in women) in 2016 [1]. In 2017, 12.5 million adults aged 20 to 79 years were diagnosed with diabetes in Brazil, the fourth among the 10 countries with the highest number of people diagnosed with diabetes in this age group [6]. According to the WHO, in Brazil, the chance of an individual aged 30 to 70 years to die of diabetes is 16.6% [1]. In 2013, the National Health Survey (NHS) estimated a self-reported diabetes prevalence of 6.2% in Brazilian adults (5.4% in men and 7.0% in women) [7]. In 2017, the Surveillance of Risk Factors and Protection for Chronic Diseases by Telephone Survey (Vigitel) conducted in Brazilian capitals found a self-reported diabetes rate of 7.6% (7.1% in men and 8.1% in women) [8].

The increase in DM prevalence is associated with the modifiable and non-modifiable factors [9]. The main non-modifiable factors associated with DM are age, ethnicity, and family history of DM [10]. However, DM is mainly caused by increased modifiable risk factors such as inadequate diet, overweight/obesity, physical inactivity, alcohol and tobacco abuse, and the presence of other conditions (hypertension and dyslipidemia) [7,11,12]. 

In 2013, the WHO established the Global Plan of Action for the Prevention and Control of Chronic Noncommunicable Diseases (NCDs), including diabetes, which aims to reduce premature mortality by these diseases and their factors by 25.0% (physical inactivity, alcohol and tobacco use, and inadequate diet) by 2025 [13]. In Brazil, the Ministry of Health launched Strategic Action Plan for Coping with NCDs in Brazil, 2011–2022, focusing on the main chronic diseases and their risk factors, establishing DM as a public health priority [14].

Considering that DM is one of the five most important diseases resulting in disease burden in Brazil [4] and that existing studies are more frequently conducted in more developed regions in the country [15], population-based epidemiological studies are essential for the evaluation and development of public healthcare policies for the prevention and control of this disease, specifically those that consider the distinctiveness of each population and subgroups who are at higher risk of developing this disease. 

Thus, this study aimed to estimate the prevalence and risk factors for self-reported DM in the adult population of the Central-West region of Brazil.

## 2. Material and Methods

### 2.1. Data Source

This was a cross-sectional, household-based study using NHS data conducted in 2013, and this study was considered as the most comprehensive nationwide population survey on risk/protection factors for NCDs conducted in Brazil [16]. The NHS was developed by the Ministry of Health, Oswaldo Cruz Foundation and the Brazilian Institute of Geography and Statistics (IBGE) and aimed to produce national data to characterize the health condition and lifestyle and access and use of healthcare services of the Brazilian population [16,17].

The NHS sampling plan was probabilistic using a three-stage cluster. The first stage consisted of the primary sampling units, which comprised the census tracts of the municipalities. The second stage consisted of secondary sampling units that corresponded to permanent private households, understood as households constructed to be used solely as housing for one or more persons. The third stage (tertiary sampling unit) consisted of individuals aged 18 years or older residing in the selected households [16,17]. Selection at all stages was performed by simple random sampling. In this study, we analyzed the data from the states located in the Central-West region of Brazil (Mato Grosso do Sul, Mato Grosso, and Goiás) and the Federal District (Figure 1).

NHS data were collected by IBGE data collection agents, supervisors, and coordinators, trained by the Ministry of Health, and interviews were recorded on portable computers called personal digital assistant [16,17]. The NHS questionnaire was validated by experts and included the collection of multiple key variables [17]. In the present study, data from the individual questionnaire were used, such as sociodemographic data, DM report, and potential risk factors.

From the total sample of the NHS (64,348 households) [7], 8803 households from the states comprising the Central-West region of Brazil were visited (Federal District, 2129; Goiás, 2831; Mato Grosso, 1827; Mato Grosso do Sul, 2016). Of these, 7982 households were interviewed, resulting in a household response rate of 90.7%. Moreover, 8795 randomly selected residents were invited to participate in the study (Federal District, 2129; Goiás, 2828; Mato Grosso, 1825; Mato Grosso do Sul, 2013), and 7519 individuals were interviewed, resulting in a response rate of 85.5% [17].

### 2.2. Variables

#### 2.2.1. Dependent Variable

Self-report DM (no or yes). Information regarding self-reported DM was obtained by asking the following question: “Were you diagnosed with diabetes by a doctor?”. Women who reported gestational DM were excluded from the prevalence estimate [7].

#### 2.2.2. Independent Variables

(i)Sociodemographic variablesThese variables included the following: age, stratified into groups of 18–29, 30–39, 40–59, or ≥60 years [18]; sex (male or female), used as a gender proxy; educational level (no education/incomplete or equivalent elementary school/complete or equivalent elementary school/incomplete high school or equivalent/complete high school or equivalent/incomplete higher education or equivalent/complete higher education or equivalent) [7]; self-declared race/color (white, black, brown, or others [yellow or indigenous]), according to the IBGE classification [19]; marital status (with or without a spouse); and area of residence (urban or rural) [17].

(ii)Eating habits.Eating habit variables included the following:
(ii.1)Recommended vegetable or fruit consumption (yes or no), defined as a daily consumption of at least 400 g of fruit and vegetables, equivalent to a minimum of 5 servings per day [20]. Information regarding vegetable or fruit consumption was assessed by asking the following questions about the daily frequency of eating vegetables, cooked vegetables, and fruits: “In general, how many times a day do you usually eat lettuce salad, tomatoes, salads, or any other vegetables or raw vegetables (once a day, once a day (lunch or dinner), twice a day (lunch and dinner), or thrice or more a day)?”; “In general, how many times a day do you usually eat cooked vegetables such as kale, carrot, chayote, eggplant, and zucchini (except potatoes, cassava, or yam) (once a day, once a day (lunch or dinner), twice a day (lunch and dinner), or thrice or more a day)?”; and “In general, how many times a day do you eat fruits (once a day, twice a day, or thrice or more a day)?” To estimate this variable, the sum of the vegetable consumption and fruit consumption variables was obtained [18].(ii.2)Information regarding the consumption of excess fat meat or skin of the chicken (yes or no), defined as a participant who reported consuming fatty meat and/or chicken including the skin [21], was obtained by asking the following questions: “When you eat red meat, do you usually (1) remove or (2) eat the excess visible fat?” and “When you eat chicken, do you usually (1) remove or (2) eat the skin of the chicken?”(ii.3)Information regarding the regular consumption of soda or artificial juice (yes or no), defined as a participant who reported consuming soda or artificial juice 5 days or more per week [21], was obtained by asking the following question: “How many days in a week do you usually drink soda (or artificial juice)?”.(ii.4)Information regarding the regular replacement of meals with snacks (yes or no), defined as a participant who reported replacing one meal (lunch or dinner) with sandwiches, salty snacks, or pizza 5 days or more during the week [22], was obtained by asking the following question: “How many days in a week do you replace your lunch or dinner meal with sandwiches, salty snacks, or pizza?”.

(iii)Tobacco and alcohol consumption.Tobacco and alcohol consumption variables included the following:
(iii.1)Tobacco use, categorized as never-smoker, former smoker, or smoker [23].(iii.2)Binge drinking (no or yes), defined as consumption of 5 or more single dose for men or 4 or more single dose for women [24]. Information regarding this variable was obtained by asking the following questions: “In the last 30 days, did you drink 5 or more drinks on a single occasion (for men)?” or “In the last 30 days, did you drink 4 or more drinks on a single occasion (for women)?”.

(iv)Nutritional status.Individuals were classified according to body mass index (BMI) as underweight (BMI <18.4 kg/m^2^), eutrophic (BMI, 18.4–24.9 kg/m^2^), overweight (BMI, 25.0–29.9 kg/m^2^), and obese (BMI ≥30 kg/m^2^) [25]. Weight and height were measured using a portable electronic scale and portable stadiometer, respectively, calibrated and used by trained collection agents to standardize measurements. Two weight and height measurements were performed in each participant. The final measure of weight and height considered was the result of the average between the two measurements. BMI was calculated by dividing weight in kilograms by height squared in meters [26].

(v)Physical inactivity (no or yes).The practice of physical activity was measured based on the participants answers on a questionnaire composed of questions about the frequency and duration of physical activity practice in different domains (leisure physical activity (AFL), displacement physical activity (AFD), and physical activity at work (AFT)). For each of these domains evaluated, physical activity practice scores were constructed by multiplying the weekly frequency by the duration time on the days the activity was performed. These questions follow the Vigitel questionnaire model, which assesses physical activity practice in leisure, work, and commuting, through a set of questions already used in international surveys and questionnaires used in the area (such as the International Physical Activity Questionnaire) [27,28]. In the present article, the participant who practiced less than 150 min per week of physical activity was considered physically inactive, considering the sum of the scores (in minutes) of AFL, AFD, and AFT [27].

(vi)Information regarding self-reported hypertension (no or yes), excluding hypertension during pregnancy [29], was obtained by asking the following question: “Has a doctor ever diagnosed you with hypertension (high blood pressure)?”.

(vii)Information regarding self-reported dyslipidemia (no or yes) [30] was obtained using the following question: “Has a doctor ever diagnosed you with dyslipidemia?”.

### 2.3. Statistical Analysis

Data were analyzed using the Stata software, version 15.0 (Stata Corp, Texas, TX, USA). Analyses were performed using complex sample routines. Initially, a descriptive analysis of the sample was performed regarding the sociodemographic, lifestyle, and other potential risk factors for DM according to sex. Pearson’s chi-squared test corrected by the study design was performed to test for differences in sex variables. Subsequently, the prevalence of self-reported DM and its 95% confidence interval (95% CI) were estimated according to the independent variables selected and for each Federation Unit (FU) of the Central-West region of Brazil. Pearson’s chi-squared test corrected by the study design was used to verify the differences in prevalence between the FUs.

To verify the risk factors for self-reported DM, a bivariate and multivariate analyses were performed. To verify the association between the dependent variable and each independent variable, the bivariate Poisson analysis was performed. Variables with *p*-value < 0.20 were selected for the Poisson regression model to adjust for potential confounding variables [31]. The results of the bivariate analysis were presented as gross prevalence ratio (PR) and respective 95% CI. The results of the multiple regression analysis were presented as adjusted prevalence ratio (aPR), standard error, and 95% CI.

Three statistical modeling methods were performed. The global model was adjusted for age, sex, educational level, consumption of red meat/chicken with excess fat, high consumption of soda and/or artificial juice, smoking, binge drinking, self-reported hypertension, self-reported dyslipidemia, physical inactivity, and nutritional status. Sensitivity analysis was also performed to verify if the associated factors differed between the sexes. The model for the group of men was adjusted by age, educational level, self-reported race/color, marital status, consumption of red meat/chicken with excess fat, high consumption of soda and/or artificial juice, smoking, binge drinking, self-reported hypertension, self-reported dyslipidemia, and nutritional status. The model for the women’s group was adjusted for age, educational level, self-reported race/color, marital status, red meat/chicken consumption with excess fat, high consumption of soda or artificial juice, smoking, binge drinking, self-reported hypertension, self-reported dyslipidemia, physical inactivity, and nutritional status.

In all analyses, variables with *p*-value < 0.05 were considered statistically significant.

### 2.4. Ethical Aspects

The NHS 2013 was approved by the National Research Ethics Commission of the National Health Council (Protocol No. 328.159/2013). All participants were informed and informed about the study and provided informed consent for inclusion in the study.

## 3. Results

### 3.1. Sample Characteristics

Table 1 describes the characteristics of the study sample stratified by sex. Regarding age, the average was 42.9 years (95% CI, 42.6–43.2), and 34.7% of the participants were aged between 40 and 59 years. Low educational level (illiterate/incomplete elementary school) was observed in 36.3% of participants. Most lived with a partner (61.8%) and were residents of the urban area (91.2%). The most prevalent risk factors for DM were: physical inactivity (47.2%); inadequate fruit/vegetable consumption (46.6%); consumption of red meat or excess chicken fat (45.7%); overweight (35.3%); obesity (22.2%); hypertension (21.2%); smoking, more specifically former smokers (16.3%); binge drinking (16.2%), and dyslipidemia (11.0%).

Significant differences were observed between sexes regarding to schooling, marital status, area of residence, recommended consumption of fruits and/or vegetables, consumption of red meat/chicken with excess fat, high consumption of soda or artificial juice, tobacco use, binge drinking, physical inactivity, hypertension, dyslipidemia, and nutritional status, as shown in Table 1.

### 3.2. Diabetes Mellitus Prevalence

The prevalence of self-reported DM in the Central-West Region was 6.5% (95% CI, 5.7–7.3). The lowest prevalence was found in the Federal District (5.8%; 95% CI, 4.7–7.1) and the highest in Mato Grosso do Sul (7.8%; 95% CI, 6.6–9.0). The prevalence in Goiás was 6.4% (95% CI, 4.5–7.0) and in Mato Grosso 6.2% (95% CI, 4.7–7.7). There was no statistically significant difference in DM prevalence among the FU (*p*-value = 0.318).

### 3.3. Risk Associated for Diabetes Mellitus

#### 3.3.1. Bivariate Analysis

Table 2, Table 3 and Table 4 describes the association between each independent variable and self-reported DM in the bivariate analysis in global sample, men and women, respectively. 

#### 3.3.2. Multiple Regression Analysis

Table 5, Table 6 and Table 7 show the regression analysis of risk factors for self-report DM.

In the global sample, it was found that age between 40–59 years (aPR, 5.06; 95% CI, 2.11–12.15) and ≥60 years (aPR, 9.57; 95% CI, 3.96–23.15), previous smoking (former smoker) (aPR, 1.35; 95% CI, 1.07–1.70), self-reported hypertension (aPR, 2.43; 95% CI, 1.83–3.24), self-reported dyslipidemia (aPR, 1.96; 95% CI, 1.55–2.47), overweight (aPR, 1.52; 95% CI, 1.11–2.09), and obesity (aPR, 2.22; 95% CI, 1.59–3.09) were independently associated with self-reported DM (Table 5).

In the male subgroup, age ≥ 60 years (aPR, 6.81; 95% CI, 2.03–22.88), self-reported hypertension (aPR, 3.02; 95% CI, 1.87–4.89), self-reported dyslipidemia (aPR, 2.60; 95% CI, 1.78–3.79), and obesity (aPR, 2.50; 95% CI, 1.48–6.44) were independently associated with self-reported DM (Table 6).

In the subgroup of women, age 30–39 years (aPR, 3.96; 95% CI, 1.42–10.89), 40–59 years (aPR, 9.80; 95% CI, 4.05–23.73), and ≥60 years (aPR, 15.94; 95% CI, 6.34–40.06), previous smoking (former smoker) (aPR, 1.37, 95% CI, 1.02–1.84), self-reported hypertension (aPR, 2.04; 95% CI, 1.44–2.89), self–reported dyslipidemia (aPR, 1.73; 95% CI, 1.27–2.35), overweight (aPR, 1.61; 95% CI, 1.09–2.37), and obesity (aPR, 2.04; 95% CI, 1.36–3.08) were independently associated with self–report DM (Table 7).

## 4. Discussion

In this study, we presented the NHS results on the prevalence of self-reported DM and its main risk factors in the Central-West region of Brazil. Factors associated with DM in the global sample were as follows: Ages 40–59 years and ≥60 years, former smoker, self-reported hypertension, self-reported dyslipidemia, overweight, and obesity. The determinants of DM between men and women showed some differences. Physical inactivity was the most prevalent risk factor in the sample, but it was not associated with DM in the global sample in the multiple regression analysis.

The NHS was a representative epidemiological survey for Brazil and its macro-regions developed between the years 2013 and 2014. The existing investigations on the survey report aggregated data for Brazil or information on the most developed macro-regions in Brazil. Thus, data on the epidemiology of DM in the Central-West region of the country had not yet been analyzed. This study shows results on the magnitude and risk factors for DM in the Central-West region, and can contribute to the implementation of effective preventive and control strategies for DM in this region to achieve the goals of the Global Action Plan for the Prevention and Control of Noncommunicable Diseases [13] and the Strategic Action Plan for Coping with NCDs in Brazil, 2011–2022 [14]. 

The prevalence of self-reported DM estimated in the Central-West region (6.5%) was similar to the prevalence according to the NHS for Brazil (6.2%) [7]. Similar values were also observed by Vigitel in 2017 (7.6%) [8]. The prevalence of DM estimated in this study was also similar to that found in other South American countries, such as Argentina (6.3%), Bolivia (6.3%), Venezuela (6.8%), Ecuador (5.5%), and Peru (6.7%). However, it was lower than estimated in Chile (9.8%), Colombia (8.4%), Paraguay (8.8%), and Uruguay (8.3%). Studies showing differences in risk factors for DM between regions and countries may explain this difference [32]. 

Consistent with other investigations, diabetes has been associated with overweight and obesity [7,9]. Obesity is the main risk factor for the development of DM. Among the various mechanisms involved in the association between overweight and diabetes, insulin resistance is the major mechanism [33]. The systematic literature review has estimated that any individual who is overweight or obese has some level of insulin resistance [34]. Insulin resistance triggered by obesity is associated with impaired pancreatic beta cell function in producing insulin due to the accumulation of fat in the pancreas; fat accumulation in the liver, leading to increased liver insulin resistance and increased glucose production; and decreased glucose uptake in the muscles [33,34]. Therefore, prevention and control of obesity is also considered a prevention strategy for the development of DM.

In this study, an association between aging and diabetes in men and women was identified, which is consistent with other investigations [7,35]. The incidence of DM is directly associated with increased life expectancy and population aging [36,37]. The association between age and DM can be explained, in addition to the accumulation of behavioral risk factors throughout life, by intrinsic changes in the aging process, such as increased insulin resistance, which in turn is associated with adiposity and physical inactivity [38]. 

Moreover, another important finding of this study is that women in the younger age group (from 30 years old) had a positive association with DM, and in the group of men, this association has not been established. That is, younger women had a higher prevalence of DM compared to men of the same age group. Although the diagnosis of the disease is more prevalent in individuals of both sexes with older age, the unfolding prevalence of this disease in younger women, considering the likely early onset of secondary complications, limitations, and decreased quality of life of these women, is worth noting. Moreover, the association between younger age and self-reported DM in women may be explained by the increased demand of women for healthcare services, which enables the early diagnosis of DM in this subgroup [15]. Also, overall, age dependency is evident in both sexes with small differences in age-specific prevalence. This result can also be explained by the distribution of overweight and obesity among men and women worldwide. According to a systematic analysis female tends to be more obese than men [39]. In Brazil, the analysis of obesity in the NHS showed that the prevalence was 16.8% among men and 24.4% among women, with women being higher than men in all age groups, including the youngest. When compared by gender, both the prevalence of overweight and obesity were higher in females. Overweight and obesity become more prevalent with increasing age in both sexes, however, in general, it tends to decrease after 60 years of age. In men, the highest prevalence of obesity is found in the 40–49 age range, and in women, in the 50–59 age range [26]. In fact, in this investigation, the prevalence of obesity was higher in women than men (25.1% versus 19.2%; Table 1).

Smoking is undoubtedly a modifiable risk factor for DM. In this study, we found a positive association between smoking cessation history and diabetes. Nicotine is known to directly alter glucose homeostasis, increasing the chance of developing DM [40]. Thus, quitting smoking is a short-term risk factor for the development of the disease. Our observation in the risk of DM after smoking cessation is consistent with a study that used data from three large cohorts showed that during an average of 19.6 years of follow-up, people who recently stopped smoking had a higher risk of type 2 diabetes than current smokers [41]. Although the weight gain commonly seen after smoking cessation may explain such a high risk of diabetes, this interruption in a short period of time can also decrease insulin sensitivity and signaling [40]. Thus, follow-up studies involving smoking cessation time and number of cigarettes smoked can better clarify this association.

Hypertension and diabetes share common risk factors such as obesity, physical inactivity, and inappropriate eating habits [42]. However, hypertension is also an independent risk factor for DM [43], which is consistent with the results of this study. As the two diseases have a close pathophysiological association, actions to prevent and control hypertension can reduce the diabetes burden reciprocally [18,42].

Dyslipidemia, obtained in the NHS by the participant’s self-report on the medical diagnosis of “high cholesterol or high triglycerides (s)”, is characterized by hypertriglyceridemia, low high-density lipoprotein fraction of cholesterol, and increased number of dense particles of low-density lipoprotein fraction [44,45]. A study that analyzed the prevalence of self-reported dyslipidemia in the Brazilian population from NHS data estimated that one in eight Brazilians refers to the medical diagnosis of high cholesterol [30]. The association between DM and dyslipidemia can be explained by the action of hyperglycemia, which alters lipid metabolism and degradation, resulting in increased free fatty acid production and circulating lipid imbalance. Thus, the high concentration of fatty acids in individuals with DM leads to insulin resistance and increased fat production, resulting in hypertriglyceridemia [44].

The present study has some limitations. First, the cross-sectional design of the study does not allow the establishment of a causal association between DM and the investigated variables (directionality bias). Additionally, the outcome of the study (medical diagnosis of DM) was self-reported, a limited measurement method that may lead to an underestimation of disease prevalence. Similarly, behavioral data were also self-reported, subject to response and memory bias by the respondents. Still, the diagnostic criterion used in this study was self-reported and does not differentiate type 1 diabetes from type 2 diabetes. Some variables, such as family history of DM, were also not investigated. Despite the limitations, this study is one of the first investigations to assess the epidemiological situation of DM in the Central-West region of Brazil, and the results presented can significantly contribute to the formulation and monitoring of public policies aimed at the promotion and surveillance of diabetes in the population in question.

## 5. Conclusions

In conclusion, the prevalence of DM was 6.5%. DM was associated with advanced age; smoking history, that is, former smokers; hypertension; dyslipidemia; overweight; and obesity, factors that also contribute to the occurrence of other noncommunicable chronic diseases. Schooling, race/skin color, eating habits, physical inactivity, and binge drinking were not factors associated with DM in the multiple regression models. Some differences in risk factors between men and women were noted. Our findings support the need to strengthen public health promotion and diabetes prevention policies in the Central-West region of Brazil. Health promotion strategies emphasizing tobacco use reduction, healthy eating programs, and physical activity performance should be targeted by health policies to reduce DM-associated comorbidities aimed at reducing the prevalence of the disease and strengthening public health.

## Figures and Tables

**Figure 1 ijerph-17-06497-f001:**
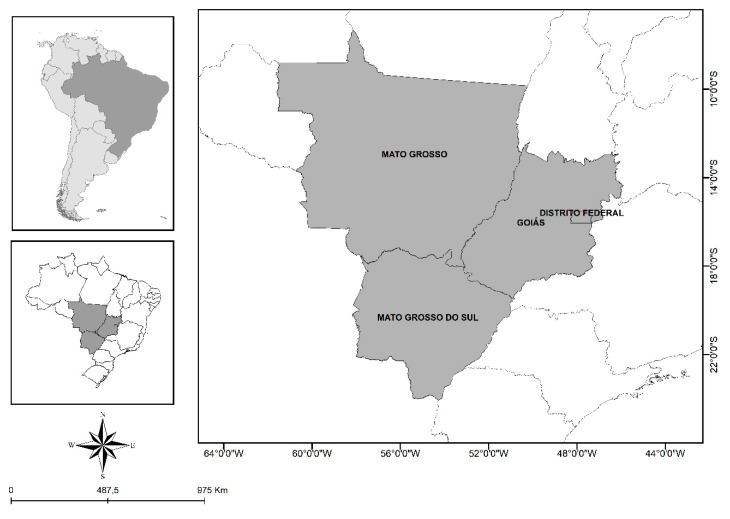
Geographic localization of study areas.

**Table 1 ijerph-17-06497-t001:** Sociodemographic characteristics, lifestyle, and nutritional status of the adult population. Central-West, Brazil, 2013.

Variables	All (*n* = 7519)		Male (*n* = 3286)		Women (*n* = 4233)		*p*-Value *
	%	95% CI	%	95% CI	%	95% CI	
**Age group (years)**							
18–29	27.8	26.4–29.2	28.6	26.2–31.1	27.0	25.3–28.9	0.366
30–39	21.9	20.7–23.2	22.4	20.5–24.3	21.5	20.0–23.1	
40–59	34.7	33.3–36.1	34.3	32.2–36.5	35.0	33.1–36.9	
≥60	15.6	14.4–16.9	14.7	13.0–16.5	16.5	15.0–18.1	
**Schooling**							
Incomplete primary education/no education	36.3	34.6–38.2	38.6	36.1–41.0	34.3	32.2–36.4	0.008
Complete primary education	16.2	15.1–17.4	16.7	15.0–18.7	15.8	14.3–17.4	
Secondary education	33.2	31.7–34.7	31.8	29.7–34.0	34.4	32.4–36.6	
Complete/incomplete higher education	14.2	13.0–15.5	12.9	11.4–14.6	15.4	13.9–17.1	
**Race/skin color**							
White	39.5	37.6–41.5	39.2	36.7–41.7	39.8	37.4–42.2	0.752
Black	7.6	6.8–8.4	8.0	6.8–9.3	7.2	6.3–8.2	
Brown	51.5	49.6–53.3	51.5	49.0–54.0	51.4	49.2–53.7	
Others	1.4	1.3–1.8	1.3	0.9–2.0	1.5	1.1–2.1	
**Marital status**							
Without partner	38.2	36.7–39.8	36.0	33.5–38.5	40.3	38.4–42.1	0.009
With partner	61.8	60.2–63.3	64.0	61.5–66.5	59.7	57.9–61.5	
**Area of residence**							
Urban	91.2	90.6–91.8	90.4	89.4–91.4	91.9	90.9–92.8	0.048
Rural	8.8	8.2–9.4	9.6	8.6–10.6	8.1	7.2–9.1	
**Recommended consumption of fruits and/or vegetables**							
No	46.6	42.7–46.5	46.9	44.3–49.6	42.4	40.2–44.7	0.004
Yes	55.4	53.5–57.3	53.1	50.4–55.7	57.6	55.3–59.8	
**Consumption of red meat/excess fat chicken**							
No	54.3	52.5–56.0	44.6	42.1–47.2	63.0	60.9–65.1	<0.001
Yes	45.7	44.0–47.5	55.4	52.8–57.9	37.0	34.9–39.1	
**High consumption of soda or artificial juice**							
No	72.3	70.6–74.0	67.7	65.0–70.4	77.5	74.7–78.2	<0.001
Yes	27.7	26.0–29.4	32.3	29.6–35.0	23.4	21.8–25.3	
**Replaces meals with snacks**							
No	92.5	91.4–93.4	92.3	90.8–93.5	92.7	91.5–93.7	0.603
Yes	7.5	6.6–8.5	7.7	6.5–9.2	7.3	6.3–8.5	
**Tobacco use**							
Never-smoker	70.2	68.6–71.8	63.6	61.2–65.8	76.3	74.4–78.1	<0.001
Former smoker	16.3	15.1–17.6	19.6	17.9–21.6	13.3	12.0–14.7	
Smoker	13.4	12.5–14.5	16.8	15.1–18.6	10.4	9.2–11.7	
**Binge drinking**							
No	83.8	82.5–85.1	76.0	73.9–78.0	91.0	89.7–92.1	<0.001
Yes	16.2	14.9–17.4	24.0	22.0–26.1	9.0	7.9–10.3	
**Physical inactivity**							
No	52.8	51.2–54.4	59.6	57.3–61.8	46.6	44.6–48.7	<0.001
Yes	47.2	45.6–48.8	40.4	38.2–42.7	53.4	51.3–55.4	
**Hypertension**							
No	78.8	77.4–80.0	81.6	79.5–83.4	76.2	74.5–77.8	<0.001
Yes	21.2	20.0–22.5	18.4	16.6–20.4	23.8	22.1–25.5	
**Dyslipidemia**							
No	89.0	88.0–90.0	92.5	91.2–93.6	85.8	84.4–87.2	<0.001
Yes	11.0	10.0–12.0	7.5	6.4–8.8	14.1	12.8–15.6	
**Nutritional status**							
Low weight	2.1	1.7–2.6	1.7	1.2–2.4	2.5	1.9–3.2	<0.001
Eutrophic	40.3	38.8–41.8	40.6	38.2–43.0	40.1	38.2–41.9	
Overweight	35.3	33.8–36.9	38.5	36.0–41.1	32.3	30.5–34.2	
Obesity	22.2	21.0–23.5	19.2	17.4–21.0	25.1	23.5–26.9	

95% CI: 95% Confidence Interval; * Pearson’s chi-square test corrected by study design.

**Table 2 ijerph-17-06497-t002:** Prevalence of self-report diabetes mellitus according to sociodemographic characteristics, lifestyle, and nutritional status in the adult population. Central-West, Brazil, 2013.

Variables	Total	Diabetes Mellitus		PR (95% CI)	*p*-Value *
		*n* = 492	% (95% CI)		
**Age group (years)**					
18–29	1797	12	0.6 (0.3–1.5)	1.00	
30–39	1758	32	2.3 (1.5–3.6)	3.58 (1.42–9.03)	0.007
40–59	2698	209	7.4 (6.3–8.7)	11.44 (4.96–26.40)	<0.001
≥60	1266	239	20.4 (19.7–24.3)	31.46 (13.59–72.85)	<0.001
**Sex**					
Male	3386	178	5.3 (4.3–6.5)	1.00	
Female	4233	314	7.5 (6.5–8.7)	1.43 (1.11–1.82)	0.004
**Schooling**					
Incomplete primary education/no education	1172	58	4.6 (3.4–6.2)	1.00	
Complete primary education	2393	84	3.3 (2.5–4.3)	0.73 (0.50–1.07)	0.106
Secondary education	1177	61	4.9 (3.6–6.6)	1.07 (0.69–1.66)	0.764
Complete/incomplete higher education	2787	289	10.8 (9.2–12.5)	2.35 (1.70–3.29)	<0.001
**Race/skin color**					
White	3007	197	6.6 (5.4–8.1)	1.00	
Black	623	53	8.2 (6.0–11.0)	1.25 (0.86–1.80)	0.237
Brown	3768	232	6.0 (5.1–7.1)	0.91 (0.71–1.19)	0.522
Others	120	10	8.6 (3.9–17.6)	1.30 (1.59–2.83)	0.511
**Marital status**					
Without partner	3229	245	6.6 (5.6–7.7)	1.00	
With partner	4290	247	6.4 (5.4–7.5)	0.97 (0.77–1.21)	0.775
**Area of residence**					
Urban	1052	66	6.5 (5.7–7.3)	1.00	
Rural	6467	426	6.4 (4.9–8.4)	1.00 (0.74–1.35)	0.977
**Recommended consumption of fruits and/or vegetables**					
No	3226	188	6.1 (5.0–7.5)	1.00	
Yes	4293	304	6.7 (5.8–7.8)	1.10 (0.86–1.40)	0.461
**Consumption of red meat/excess fat chicken**					
No	4166	320	7.7 (6.6–8.9)	1.00	
Yes	353	172	5.0 (0.4–6.1)	0.65 (0.50–0.84)	0.001
**High consumption of soda or artificial juice**					
No	5570	420	7.4 (6.5–8.3)	1.00	
Yes	1949	72	4.1 (2.9–5.7)	0.55 (0.39–0.79)	0.001
**Replaces meals with snacks**					
No	6977	464	6.6 (5.8–7.4)	1.00	
Yes	552	28	5.4 (3.3–8.5)	0.82 (0.50–1.32)	0.416
**Tobacco use**					
Never-smoker	5226	284	5.0 (4.3–5.8)	1.00	
Former smoker	1227	153	13.9 (11.1–17.2)	2.66 (2.13–3.59)	<0.001
Smoker	1066	55	4.9 (3.5–6.8)	0.97 (0.69–1.38)	0.888
**Binge drinking**					
No	6375	459	7.0 (6.2–8.0)	1.00	
Yes	1144	33	3.4 (2.0–5.5)	0.48 (0.28–0.80)	0.006
**Physical inactivity**					
No	3941	206	5.3 (4.3–6.4)	1.00	
Yes	3578	286	7.8 (6.7–9.1)	1.48 (1.15–1.90)	0.002
**Hypertension**					
No	5858	169	2.9 (2.4–3.5)	1.00	
Yes	1661	323	19.6 (16.9–22.6)	6.72 (5.31–8.49)	<0.001
**Dyslipidemia**					
No	6661	306	4.5 (3.9–5.2)	1.00	
Yes	858	186	22.2 (18.3–26.7)	4.93 (3.91–6.22)	<0.001
**Nutritional status**					
Eutrophic	2857	97	2.8 (2.2–3.6)	1.00	
Low weight	151	6	2.9 (1.2–6.9)	1.03 (0.40–2.59)	0.956
Overweight	2648	182	7.1 (5.8–8.7)	2.57 (1.86–3.54)	<0.001
Obesity	1666	200	12.7 (10.7–15.0)	4.57 (3.35–6.22)	<0.001

PR: Crude Prevalence Ratio; 95% CI: 95% Confidence Interval; * Wald’s chi-square test.

**Table 3 ijerph-17-06497-t003:** Prevalence of self-report diabetes mellitus according to sociodemographic characteristics, lifestyle, and nutritional status in men. Central-West, Brazil, 2013.

Variables	Total	Diabetes Mellitus		PR (95% CI)	*p*-Value *
		*n* = 178	% (95% CI)		
**Age group (years)**					
18–29	784	5	0.9 (0.3–2.8)	1.00	
30–39	759	11	2.4 (1.2–4.8)	2.60 (0.70–9.76)	0.154
40–59	1220	76	5.7 (4.3–7.6)	6.22 (1.97–19.65)	0.002
≥60	523	86	17.1 (12.7–22.7)	18.73 (5.82–60.29)	<0.001
**Schooling**					
Incomplete primary education/no education	484	24	4.7 (2.9–7.3)	1.00	
Complete primary education	988	34	3.6 (2.3–5.4)	0.76 (0.41– 1.44)	0.406
Secondary education	516	27	4.9 (3.0–7.9)	1.06 (0.54–2.07)	0.864
Complete/incomplete higher education	1298	93	7.0 (5.2–9.5)	1.51 (0.88–2.59)	0.136
**Race/skin color**					
White	1294	76	6.3 (4.5–8.7)	1.00	
Black	292	20	6.5 (3.9–10.5)	1.03 (0.57–1.88)	0.919
Brown	1641	80	4.2 (3.2–5.5)	0.67 (0.44–1.03)	0.067
Others	49	2	8.1 (1.8–29.3)	1.29 (0.30–5.50)	0.732
**Marital status**					
Without partner	1308	65	3.3 (2.3–4.6)	1.00	
With partner	1978	113	6.4 (5.0–8.2)	1.96 (1.26–3.03)	0.003
**Area of residence**					
Urban	540	31	4.7 (3.1–7.1)	1.00	
Rural	2746	147	5.5 (4.3–6.6)	1.12 (0.70–1.78)	0.614
**Recommended consumption of fruits and/or vegetables**					
No	1476	69	5.1 (3.5–7.2)	1.00	
Yes	1810	109	5.4 (4.3–6.8)	1.07 (0.70–1.64)	0.749
**Consumption of red meat/excess fat chicken**					
No	1464	106	7.0 (5.4–9.0)	1.00	
Yes	1822	72	3.9 (2.8–5.4)	0.56 (0.36–0.85)	0.007
**High consumption of soda or artificial juice**					
No	2314	147	5.9 (4.7–7.3)	1.00	
Yes	972	31	4.0 (2.3–6.7)	0.67 (0.38–1.20)	0.178
**Replaces meals with snacks**					
No	3054	167	5.3 (4.3–6.6)	1.00	
Yes	232	11	4.3 (2.0–9.0)	0.81 (0.37–1.77)	0.597
**Tobacco use**					
Never-smoker	2005	86	3.9 (3.0–5.1)	1.00	
Former smoker	650	70	11.5 (8.2–16.0)	2.92 (1.90–4.5)	<0.001
Smoker	631	22	3.0 (1.8–4.9)	0.76 (0.43–1.35)	0.352
**Binge drinking**					
No	2505	156	5.8 (4.7–7.2)	1.00	
Yes	781	22	3.5 (1.8–6.4)	0.59 (0.30–1.15)	0.120
**Physical inactivity**					
No	1955	88	4.7 (3.5–6.3)	1.00	
Yes	1331	90	6.1 (4.6–8.1)	1.29 (0.86–1.95)	0.216
**Hypertension**					
No	2670	68	2.4 (1.7–3.3)	1.0	
Yes	616	110	18.0 (13.8–23.2)	7.54 (4.93–11.51)	<0.001
**Dyslipidemia**					
No	3018	120	3.8 (3.0–4.7)	1.00	
Yes	268	58	23.8 (16.3–33.4)	6.31 (4.11–9.69)	<0.001
**Nutritional status**					
Eutrophic	1293	41	2.5 (1.6–3.7)	1.00	
Low weight	51	3	4.1 (1.0–14.8)	1.65 (0.40–6.90)	0.488
Overweight	1283	70	5.1 (3.5–7.3)	2.07 (1.20–3.59)	0.009
Obesity	619	64	11.6 (8.5–15.6)	4.70 (2.77–7.98)	<0.001

PR: Crude Prevalence Ratio; 95% CI: 95% Confidence Interval; * Wald’s chi-square test.

**Table 4 ijerph-17-06497-t004:** Prevalence of self-report diabetes mellitus according to sociodemographic characteristics, lifestyle, and nutritional status in women. Central-West, Brazil, 2013.

Variables	Total	Diabetes Mellitus		PR (95% CI)	*p*-Value *
		*n* = 314	% (95% CI)		
**Age group (years)**					
18–29	1013	7	0.4 (0.2–0.9)	1.00	
30–39	999	21	2.3 (1.3–3.8)	5.57 (2.14–15.55)	0.001
40–59	1478	133	9.0 (7.4–10.9)	22.90 (9.78–53.64)	<0.001
≥60	743	153	23.1 (19.0–27.7)	58.91 (25.30–137.20)	<0.001
**Schooling**					
Incomplete primary education/no education	688	34	4.5 (2.9–6.8)	1.00	
Complete primary education	1405	50	3.1 (2.2–4.4)	0.69 (0.43–1.12)	0.132
Secondary education	651	34	4.8 (3.2–7.2)	1.07 (0.59–1.96)	0.813
Complete/incomplete higher education	1489	196	14.6 (12.3–17.2)	3.23 (2.05–5.09)	<0.001
**Race/skin color**					
White	1713	121	6.8 (5.3–8.8)	1.00	
Black	331	33	9.9 (6.7–14.5)	1.45 (0.92–2.29)	0.107
Brown	2117	152	7.7 (6.3–9.4)	1.12 (0.80–1.58)	0.503
Others	71	8	8.9 (4.0–18.7)	1.30 (0.57–2.98)	0.531
**Marital status**					
Without partner	1921	180	9.3 (7.8–11.1)	1.00	
With partner	2312	134	6.3 (5.1–7.8)	0.68 (0.52–0.89)	0.006
**Area of residence**					
Urban	512	35	8.3 (5.7–11.8)	1.00	
Rural	3721	279	7.5 (6.4–8.7)	0.90 (0.61–1.33)	0.614
**Recommended consumption of fruits and/or vegetables**					
No	1750	119	7.2 (5.8–8.9)	1.00	
Yes	2483	195	7.8 (6.5–9.3)	1.08 (0.82–1.44)	0.573
**Consumption of red meat/excess fat chicken**					
No	2702	214	8.1 (6.8–9.6)	1.00	
Yes	1531	100	6.6 (5.1–8.5)	0.81 (0.59–1.11)	0.189
**High consumption of soda or artificial juice**					
No	3256	273	8.6 (7.4–9.9)	1.00	
Yes	977	41	4.2 (2.7–6.4)	0.49 (0.31–0.77)	0.002
**Replaces meals with snacks**					
No	3913	297	7.6 (6.6–8.8)	1.00	
Yes	320	17	6.3 (3.5–11.3)	0.83 (0.46–1.51)	0.546
**Tobacco use**					
Never-smoker	3221	198	5.8 (4.9–6.9)	1.00	
Former smoker	577	171	17.1 (12.9–22.3)	2.92 (2.10–4.07)	<0.001
Smoker	435	33	7.7 (5.0–11.6)	1.31 (0.84–2.06)	0.228
**Binge drinking**					
No	3870	303	8.0 (6.9–9.2)	1.00	
Yes	363	11	3.1 (1.4–6.9)	0.39 (0.17–0.89)	0.026
**Physical inactivity**					
No	1986	118	5.9 (4.7–7.5)	1.00	
Yes	2247	196	9.0 (7.5–10.6)	1.51 (1.13–2.03)	0.006
**Hypertension**					
No	3188	101	3.4 (2.7–4.4)	1.00	
Yes	1045	213	20.7 (17.6–24.1)	6.04 (4.50–8.10)	<0.001
**Dyslipidemia**					
No	3643	186	5.2 (4.4–6.3)	1.00	
Yes	590	128	21.5 (17.4–26.1)	4.10 (3.13–5.38)	<0.001
**Nutritional status**					
Eutrophic	1564	59	3.1 (2.3–4.2)	1.00	
Low weight	100	3	2.1 (0.6–7.0)	0.67 (0.18–2.41)	0.536
Overweight	1365	112	9.4 (7.4–11.8)	3.05 (2.04–4.55)	<0.001
Obesity	1047	136	13.5 (11.0–16.5)	4.38 (2.99–6.40)	<0.001

PR: Crude Prevalence Ratio; 95% CI: 95% Confidence Interval; * Wald’s chi-square test.

**Table 5 ijerph-17-06497-t005:** Multiple regression analysis of risk factors for self-report diabetes mellitus in the adult population. Central-West, Brazil, 2013.

Variables	aPR	95% CI	Standard Error	*p*-Value *
**Age group (years)**				
18–29	1.00			
30–39	2.56	1.00–6.56	1.23	0.050
40–59	5.06	2.11–12.15	2.26	<0.001
≥60	9.57	3.96–23.15	4.31	<0.001
**Sex**				
Male	1.00			
Female	1.07	0.84–1.36	0.13	0.575
**Schooling**				
Incomplete primary education/no education	1.00			
Complete primary education	1.03	0.69–1.54	0.21	0.876
Secondary education	1.19	0.76–1.86	0.27	0.444
Complete/incomplete higher education	1.27	0.91–1.77	0.21	0.155
**Consumption of red meat/excess fat chicken**				
No	1.00			
Yes	0.88	0.70–1.10	0.10	0.256
**High consumption of soda or artificial juice**				
No	1.00			
Yes	0.97	0.72–1.33	0.15	0.876
**Tobacco use**				
Never–smoker	1.00			
Former smoker	1.35	1.07–1.70	0.16	0.012
Smoker	1.08	0.78–1.51	0.18	0.638
**Binge drinking**				
No	1.00			
Yes	0.81	0.52–1.27	0.18	0.364
**Physical inactivity**				
No	1.00			
Yes	1.04	0.82–1.31	0.12	0.742
**Hypertension**				
No	1.00			
Yes	2.43	1.83–3.24	0.35	<0.001
**Dyslipidemia**				
No	1.00			
Yes	1.96	1.55–2.47	0.23	<0.001
**Nutritional status**				
Eutrophic	1.00			
Low weight	1.03	0.44–2.42	0.45	0.951
Overweight	1.52	1.11–2.09	0.24	0.009
Obesity	2.22	1.59–3.09	0.38	<0.001

aPR: Adjusted Prevalence Ratio; 95% CI: 95% Confidence Interval; * Wald’s chi-square test.

**Table 6 ijerph-17-06497-t006:** Multiple regression analysis of risk factors for self-report diabetes mellitus in men. Central-West, Brazil, 2013.

Variables	aPR	95% CI	Standard Error	*p*-Value *
**Age group (years)**				
18–29	1.00			
30–39	2.00	0.55–7.27	1.32	0.287
40–59	2.92	0.89–9.58	1.77	0.077
≥60	6.81	2.03–22.88	4.20	0.002
**Schooling**				
Incomplete primary education/no education	1.00			
Complete primary education	1.43	0.75–2.73	0.47	0.279
Secondary education	1.64	0.83–3.525	0.57	0.155
Complete/incomplete higher education	1.18	0.69–2.03	0.33	0.546
**Race/skin color**				
White	1.00			
Black	1.10	0.62–1.93	0.32	0.749
Brown	0.76	0.52–1.09	0.14	0.136
Others	0.97	0.35–2.67	0.50	0.960
**Marital status**				
Without partner	1.00			
With partner	1.05	0.71–1.55	0.20	0.817
**Consumption of red meat/excess fat chicken**				
No	1.00			
Yes	0.71	0.49–1.03	0.14	0.075
**High consumption of soda or artificial juice**				
No	1.00			
Yes	1.02	0.64–1.62	0.24	0.940
**Tobacco use**				
Never–smoker	1.00			
Former smoker	1.23	0.85–1.78	0.23	0.282
Smoker	0.93	0.53–1.62	0.26	0.791
**Binge drinking**				
No	1.00			
Yes	0.68	0.40–1.15	0.18	0.151
**Hypertension**				
No	1.00			
Yes	3.02	1.87–4.89	0.74	<0.001
**Dyslipidemia**				
No	1.00			
Yes	2.60	1.78–3.79	0.50	<0.001
**Nutritional status**				
Eutrophic	1.00			
Low weight	1.70	0.45–6.44	1.15	0.431
Overweight	1.35	0.79–2.29	0.36	0.269
Obesity	2.50	1.48–6.44	0.67	0.001

aPR: Adjusted Prevalence Ratio; 95% CI: 95% Confidence Interval; * Wald’s chi-square test.

**Table 7 ijerph-17-06497-t007:** Multiple regression analysis of risk factors for self–report diabetes mellitus in women. Central–West, Brazil, 2013.

Variables	aPR	95% CI	Standard Error	*p*–Value *
**Age group (years)**				
18–29	1.00			
30–39	3.94	1.42–10.89	2.04	0.008
40–59	9.80	4.05–23.73	4.41	<0.001
≥60	15.94	6.34–40.06	7.48	<0.001
**Schooling**				
Incomplete primary education/no education	1.00			
Complete primary education	0.85	0.52–1.37	0.21	0.494
Secondary education	0.93	0.51–1.70	0.29	0.815
Complete/incomplete higher education	1.30	0.83–2.07	0.30	0.252
**Race/skin color**				
White	1.00			
Black	1.44	0.94–2.19	0.31	0.094
Brown	1.19	0.88–1.62	0.19	0.266
Others	1.12	0.46–2.68	0.50	0.809
**Marital status**				
Without partner	1.00			
With partner	0.80	0.60–1.06	0.11	0.121
**Consumption of red meat/excess fat chicken**				
No	1.00			
Yes	1.07	0.82–1.39	0.15	0.632
**High consumption of soda or artificial juice**				
No	1.00			
Yes	0.88	0.56–1.36	0.20	0.552
**Tobacco use**				
Never–smoker	1.00			
Former smoker	1.37	1.02–1.84	0.21	0.035
Smoker	1.22	0.80–1.87	0.26	0.351
**Binge drinking**				
No	1.00			
Yes	0.90	0.40–2.03	0.37	0.803
**Physical inactivity**				
No	1.00			
Yes	1.12	0.86–1.46	0.15	0.391
**Hypertension**				
No	1.00			
Yes	2.04	1.44–2.89	0.36	<0.001
**Dyslipidemia**				
No	1.00			
Yes	1.73	1.27–2.35	0.27	0.001
**Nutritional status**				
Eutrophic	1.00			
Low weight	0.66	0.19–2.29	0.42	0.512
Overweight	1.61	1.09–2.37	0.32	0.016
Obesity	2.04	1.36–3.08	0.43	<0.001

aPR: Adjusted Prevalence Ratio; 95% CI: 95% Confidence Interval; * Wald’s chi-square test.

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
