# Peer review of "Prevalence and Risk Factors for Self-Report Diabetes Mellitus: A Population-Based Study"

_ijerph, 2020, doi:10.3390/ijerph17186497_

Round 1
Reviewer 1 Report
The authors present a cross-sectional analysis of cohort data from Brazil, investigating prevalence and risk factors for self-reported T2DM.
The introduction provides sufficient information to lead to the actual study.
Methods are described mostly accurately.
Smoking status should be consistently labelled as "never-smokers" rather than "non-smokers" (line 135) as it was done in the tables.
If anthropometry was done by actual measurements, it is unclear, why prevalence of diabetes was assessed by self report, only. It should be clarified, why measurements of fasting glucose, oGTT and / or HbA1c were not done.
Questions for hypertension and/or dyslipidemia should also include the aspect of medication. Some patients are unaware of the indication of their pharmaceutical treatment, thereby having a diagnosed hypertension or dyslipidemia, without knowing it.
Definition for alcohol abuse is quite irregular. Under the used criteria, subjects with daily intake of up to 3 / 4 drinks would not be considered to have a problematic alcohol consumption. Consistent labelling as "binge drinking" throughout the manuscript will solve the inconsistency between the label "alcohol abuse" (in general) and "binge drinking" (as one form of alcohol abuse). However, it leaves the issue, that most cases of problematic alcohol intake (drinkers with daily supranormal intake) are not identified properly.
Analyses for continuous parameters (age, BMI) should be extended to a graphic exploration and suitable linear / non-linear regression analysis of smaller age and BMI intervals in order to provide a more detailed insight into the mathematical relation and potential cut-offs for elevated risk.
Results:
Residence, snacking and fruits/vegetables appear in the supplementary materials, but not in the main part. Please explain.
Please also use the same order for all factors in all tables for better comparability.
In all tables, low body weight should be used a separate category (and not be integrated together with the reference group "normal body weight"). Patients with SIDD or MARD (according to Ahlqvist et al. 2018) are often normal- to underweight.
Table 3 lacks the information on physical activity. Please complete the table and - if necessary - re-do the analysis.
Line 302 ff. reports, that active smoking is associated with T2DM which is not supported by the presented data. Please clarify.
Also, the authors should extend their discussion on why smoking cessation is a stronger risk factor than smoking itself.
The discussion also does not cover findings from the supplementary sectioni, e.g. on partnership, physical activity or on nutritional factors (high fat intake, high intake of sodas and artificial juices, binge drinking). These factors seem to be "protective" (supplemental data). Please clarify, why high intake of fat, alcohol and sodas should protect from T2DM.
There is also a U-shaped relation between education and T2DM risk apparant in suppl. table 1. Please explain this finding in the discussions section.
Higher T2DM in younger age for women than men is quite uncommon, please explain further.
As some factors seem to be differentially associated with T2DM risk in men and women, an additional pin-pointed interaction analysis for these factors should be done (ethnicity, education, marital status, fat intake, smoking) and discussed.
Author Response
1-Comment: The authors present a cross-sectional analysis of cohort data from Brazil, investigating prevalence and risk factors for self-reported T2DM. The introduction provides sufficient information to lead to the actual study.
Answer: Thanks for the careful evaluation of the manuscript. We sent the reformulated manuscript according to suggestions and answers to the questions below.
2- Comment: Methods are described mostly accurately.
Answer: Thanks for the feedback.
3-Comment: Smoking status should be consistently labelled as "never-smokers" rather than "non-smokers" (line 135) as it was done in the tables.
Answer: Thanks for the observation. The categories used in the text and tables have been replaced by: never-smoker, former smoker and, observation.
4-Comment: If anthropometry was done by actual measurements, it is present, why prevalence of diabetes was assessed by self-report, only. It should be clarified, why measurements of fasting glucose, oGTT and/or HbA1c were not done.
Answer: Thanks for the comment. For this manuscript, unfortunately, we were only able to analyze the self-report and did not use laboratory measures to diagnose diabetes. We recognize this point in the limitations of the study, as presented in the manuscript in the “discussion” section:
“Additionally, the outcome of the study (medical diagnosis of DM) was self-reported, a limited measurement method that may lead to na underestimation of disease prevalence.”
5- Comment: Questions for hypertension and/or dyslipidemia should also include the aspect of medication. Some patients are unaware of the indication of their pharmaceutical treatment, thereby having a diagnosed hypertension or dyslipidemia, without knowing it.
Answer: Thanks for watching. In NHS, one of the diagnostic criteria used to find out if people had hypertension was the disease referred to and diagnosed by a doctor ("Has any doctor ever told you that you have high blood pressure (high blood pressure)?" ). In addition, the questionnaire also contained questions about the use of antihypertensive drugs: "In the past two weeks, did you take any medications because of high blood pressure (high blood pressure)?". However, the participant would only answer the question about taking medications because of high blood pressure if he answered “Yes” to the question about whether a doctor has already given the diagnosis of high blood pressure. Furthermore, in the questionnaire, there were no questions about which drugs were used for hypertension. Self-reported arterial hypertension can be used as a valid population estimate, according to studies already carried out.
In the case of questions about dyslipidemia, the medication aspects for dyslipidemia referred only if a doctor or health professional had given the recommendation to take medications because of high cholesterol, so the dichotomous analysis was also performed for the diagnosis of cholesterol high.
(References: - Malta DC, Gonçalves RPF, Machado IE, Freitas MIF, Azeredo C, Szwarcwald CL. Prevalence of arterial hypertension according to different diagnostic criteria, National Health Survey. Rev Bras Epidemiol [Internet]. 2018 11 ago 2020. 21(SUPPL 1): E180021.supl.1. Avaiable from: https://www.scielo.br/pdf/rbepid/v21s1/en_1980-5497-rbepid-21-s1-e180021.pdf.
- Fontanelli MM, Teixeira JA, Sales CH, Castro MA, Cesar CLG, Alves MCGP et al. Validation of self-reported diabetes in a representative sample of São Paulo city. Rev Saúde Pública [Internet]. 2017 11 ago 2020. 51:20. Avaiable from: https://www.scielo.br/pdf/rsp/v51/0034-8910-rsp-S1518-87872017051006378.pdf).
Therefore, we were unable to assess those people who used medication, but did not report a medical diagnosis of hypertension and dyslipidemia.
6- Comment: Definition for alcohol abuse is quite irregular. Under the used criteria, subjects with daily intake of up to 3 / 4 drinks would not be considered to have a problematic alcohol consumption. Consistent labelling as "binge drinking" throughout the manuscript will solve the inconsistency between the label "alcohol abuse" (in general) and "binge drinking" (as one form of alcohol abuse). However, it leaves the issue, that most cases of problematic alcohol intake (drinkers with daily supranormal intake) are not identified properly.
Answer: thanks for the observation. We apologize for the duplicity of terms. We do not investigate addiction or other forms of alcohol abuse. The indicator used was binge drinking, defined as consumption of 5 or more single dose for men or 4 or more single dose for women. Information regarding this variable was obtained by asking the following questions: “In the last 30 days, did you drink 5 or more drinks on a single occasion (for men)?” or “In the last 30 days, did you drink 4 or more drinks on a single occasion (for women)?”. Thus, we made the correction of the term in the manuscript.
7-Comment: Analyses for continuous parameters (age, BMI) should be extended to a graphic exploration and suitable linear/non-linear regression analysis of smaller age and BMI intervals in order to provide a more detailed insight into the mathematical relation and potential cut-offs for elevated risk.
Answer: thanks for the remark. As the objective of the study was to verify how the different cutoff points of nutritional status (eutrophic, underweight, obesity and overweight) influenced obesity, this relationship is not included in the manuscript. Since it did not encompass the objective, we asked the reviewer's permission to maintain the current analyzes, except for the separation of the low weight from the eutrophic category.
8-Comment: Residence, snacking and fruits/vegetables appear in the supplementary materials, but not in the main part. Please explain.
Answer: thanks for the observation. The supplementary tables show the results of the bivariate analysis (not adjusted) and the tables that were in the body of the text, the results of the multiple regression analysis (adjusted). However, in order not to confuse the reader, we have included the supplementary tables (Table 1, Table 2 and, Table 3) that contain these variables in the body of the text.
9-Comment: Please also use the same order for all factors in all tables for better comparability.
Answer: thanks for the observation. In Table 1, which presents the descriptive analysis by sex, the variables were presented in order: age group, schooling, race / skin color, marital status, residence, recommended consumption of fruits and / or vegetables. Consumption of red meat / excess fat chicken, high consumption of soda or artificial juice, replaces meals with snacks, tobacco use, binge drinking, physical inactivity, hypertension, dyslipidemia and, nutritional status. Tables 2, 3 and 4, which represent the bivariate analysis, were presented in the same sequence as above, except in Table 2 (total sample) that has sex as an independent variable. Tables 5, 6 and 7 the primary order of the variables is the same. What differentiates are the different models. For example, in Table 7 (female), physical inactivity was included in the model, as it presented p-value <0.20 in the bivariate analysis, which did not occur in the group of men.
10-Comment: In all tables, low body weight should be used a separate category (and not be integrated together with the reference group "normal body weight"). Patients with SIDD or MARD (according to Ahlqvist et al. 2018) are often normal- to underweight.
Answer: Thanks for the observation. We agree with the reviewer. The categories were separated and new bivariate and multiple regression analyzes were performed, considering this new categorization.
11- Comment: Table 3 lacks the information on physical activity. Please complete the table and - if necessary - re-do the analysis.
Answer: we used a cutoff point of p-value<0.20 in the bivariate analysis for inclusion in the regression model (Statistical analysis in methods). As in the men's group, this did not occur, the model of this subgroup did not include this variable.
12- Comment: Line 302 ff. reports, that active smoking is associated with T2DM which is not supported by the presented data. Please clarify.
Answer: Thanks for observation. In fact, in our study, only past smoking was associated. We made the necessary correction in the manuscript.
13- Comment: Also, the authors should extend their discussion on why smoking cessation is a stronger risk factor than smoking itself.
Answer: Thanks for the observation. We included a discussion of the association between smoking cessation and T2DM.
14- Comment: -The discussion also does not cover findings from the supplementary section, e.g. on partnership, physical activity or on nutritional factors (high fat intake, high intake of sodas and artificial juices, binge drinking). These factors seem to be "protective" (supplemental data). Please clarify, why high intake of fat, alcohol and sodas should protect from T2DM.
Answer: Thanks for observation. We inform that, despite protective factors in the bivariate analysis, this association was not maintained in the multiple regression models (Tables 5, 6 and 7 in Results). Therefore, the conclusion in our manuscript was that eating habits and binge drinking were not factors associated with T2DM in the multiple regression model, so we did not add this discussion. We added this fact in the concluding session.
15- Comment-There is also a U-shaped relation between education and T2DM risk apparant in suppl. table 1. Please explain this finding in the discussions section.
Answer: Thanks for observation. We inform that this variable did not present statistical significance in our analysis (Tables 5, 6 and 7 in Results). Therefore, the conclusion in our manuscript was that schooling was not an associated factor with T2DM in the multiple regression model, so we did not add this discussion. We added this fact in the concluding session.
16- Comment: Higher T2DM in younger age for women than men is quite uncommon, please explain further.
Answer: Thanks for the observation. We included this discussion in our manuscript of section Discussion.
17- Comment:- As some factors seem to be differentially associated with T2DM risk in men and women, an additional pin-pointed interaction analysis for these factors should be done (ethnicity, education, marital status, fat intake, smoking) and discussed.
Answer: Thank you for that important observation. Interactions between sex and other variables were tested in the models. We did not find significant interactions that made it possible to maintain the regression models. The results showed the following p values for interaction
Sex and race o = 0.990.
Sex and marital status p = 0.137.
Sex and physical inactivity = 0.527
Sex and nutritional status = 0.378
Sex and education = 0.809
Sex and smoking = 1,000
Sex and consumption of meat and chicken with excess fat = 0.173
Sex and soda consumption = 0.406.
Best regards
Reviewer 2 Report
This is a traditional statistical study that is well structured and well described, but there are several issues that need to be clarified.
Major Concerns
- It is unclear why the authors used data from the 2013 studies. What is the novelty and relevance of the results obtained? This must be justified.
- It is also unclear why the authors divided patients according to the indicated ages (18-29, 30-39, 40-59, or> 60 years). What is this division based on? This also needs to be justified.
- It would be more informative if the authors compare their results with the known data for Brazil and South America as a whole (if possible). It is necessary to identify the distinctive features of the Midwest region.
- The risk factors for type 1 and type 2 diabetes are similar in many ways, but there are differences. However, the authors do not differentiate between 1 and 2 types of diabetes. What is the reason for this? If possible, it is better to separate this data. Also, did the study take into account the factor of heredity?
Author Response
Comment: This is a traditional statistical study that is well structured and well described, but there are several issues that need to be clarified.
Answer: Thanks for the positive feedback. We forward the answers to the questions below.
Comment: It is unclear why the authors used data from the 2013 studies. What is the novelty and relevance of the results obtained? This must be justified.
Answer: The NHS was a representative epidemiological survey for Brazil and its macro-regions developed between the years 2013 and 2014. The existing investigations on the survey report aggregated data for Brazil or information on the most developed macro-regions in Brazil. Thus, data on the epidemiology of DM in the Midwest region of the country had not yet been analyzed. This study shows results on the magnitude and factors associated with DM in the Midwestern macro-region, and can contribute to the implementation of effective preventive and control strategies for DM in this region to achieve the goals of the Global Plan for Coping with Chronic Diseases and the Health Plan. Strategic Actions to Combat Chronic Non-Communicable Diseases in Brazil, years 2011 to 2022.
We added this comment to the discussion.
Comment: It is also unclear why the authors divided patients according to the indicated ages (18-29, 30-39, 40-59, or> 60 years). What is this division based on? This also needs to be justified.
Answer: Thanks for the comment. In this study, the independent variable age, defined as full years or assumed age, was categorized into groups of 18-29 years, 30 to 39 years, 40 to 59 years and 60 years or more based on another investigation that estimated prevalence and risk factors. risks associated with DM, and that followed this same stratification, in order to identify possible differences in the prevalence of DM and factors associated with increasing age.
Reference:
Guimaraes RA, Neto OLM, Souza MR, Cortez-Escalante JJ, Santos TAP, Rosso CFW, et al. Epidemiology of Self-Reported Diabetes Mellitus in the State of Maranhão, Northeastern Brazil: Results of the National Health Survey, 2013. International Journal of Environmental Research and Public Health [Internet]. 2019 06 May 2019; 16: 1-9. Available from: https://www.ncbi.nlm.nih.gov/pmc/articles/PMC6339244/pdf/ijerph-16-00047.pdf.
Comment: It would be more informative if the authors compare their results with the known data for Brazil and South America as a whole (if possible). It is necessary to identify the distinctive features of the Midwest region.
Answer: Thanks for watching. In the discussion session, we compared the results of the prevalence of self-reported diabetes in the Midwest region with other studies conducted in Brazil. Other results from PNS Brasil also showed a prevalence of self-reported diabetes (6.2% (95% CI: 5.9-6.6) (Reference: Iser BPM, Stopa PS, Szwarcwald CL, Malta DC, Monteiro HOC et al. Self- reported diabetes prevalence in Brazil: results from National Health Survey 2013. Epidemiology and Health Services [Internet]. 2015 Aug 18, 2020; 24 (2). Available from: https://www.scielo.br/scielo.php?script = sci_arttext & pid = S2237-96222015000200305) .We also compare with other countries in South America. This discussion has been added.
Comment: The risk factors for type 1 and type 2 diabetes are similar in many ways, but there are differences. However, the authors do not differentiate between 1 and 2 types of diabetes. What is the reason for this? If possible, it is better to separate this data. Also, did the study take into account the factor of heredity?
Answer: As mentioned in the methodology section the NHS aimed to produce national data to characterize the health condition and lifestyle and access and use of healthcare services of the Brazilian population [16, 17]. Information regarding DM was obtained by asking the following question: “Were you diagnosed with diabetes by a doctor?” Women who reported gestational DM were excluded from the prevalence estimate.
Therefore, since this is a survey with a representative sample of the Brazilian population, to promote health policies in the area of chronic non-communicable diseases, the question was self-reported, and does not differentiate type 1 diabetes from type 2.
In addition, the participant was not asked about the family history of DM. Some studies were developed using the self-reported measure and it has good reliability to identify people with DM.
We agree with the reviewer that there are similar risk factors between the two types, but this is a limitation of our work and will be mentioned in the limitations section.
(References:
- Ning M, Zhang Q, Yang M. Comparison of self-reported and biomedical data on hypertension and diabetes: findings from the China Health and Retirement Longitudinal Study (CHARLS). BMJ Open [Internet]. 2016 Aug 11, 2020; 4; 6 (1): e009836. Avaiable from: https://pubmed.ncbi.nlm.nih.gov/26729390/.
- Peterson KL, Jacobs JP, Allender S, Alston LV, Nichols M. Characterizing the extent of misreporting of high blood pressure, high cholesterol, and diabetes using the Australian Health Survey. BMC Public Health [Internet]. 2016 Aug 11, 2020; 16: 695. Avaiable from: https://bmcpublichealth.biomedcentral.com/articles/10.1186/s12889-016-3389-y.
- Paalanen L, Koponen P, Laatikainen T, Tolonen H. Public health monitoring of hypertension, diabetes and elevated cholesterol: comparison of different data sources. Eur J Public Health [Internet]. 2018 Aug 11, 2020; 1; 28 (4): 754-765.Avaiable from: https://pubmed.ncbi.nlm.nih.gov/29462296/.
- Pastorino S, Richards M, Hardy Rebecca, Abington Jane, Wills Andrew et al. Validation of self-reported diagnosis of diabetes in the 1946 British birth cohort. 2015 Aug 11, 2020; 9 (5): 397–400. Avaiable from: https://www.ncbi.nlm.nih.gov/pmc/articles/PMC4582042/.
Reviewer 3 Report
In this study, the authors examined the prevalence and risk factors for self-reported diabetes in the adult population of the Midwest region of Brazil. Using NHS data conducted in 2013, 7,519 adults were included in the study. The prevalence of diabetes was 6.5%. In the global sample, age, smoking, hypertension, dyslipidemia, overweight and obesity were independently associated with self-reported diabetes.
Overall, the use of large cohort was a strength of the study, however, there are critiques as described below.
- The findings of study were mostly confirmative and novelty of the study was not clear. The authors need to describe novelty of the study more precisely, or the manuscript may be suited to publish in more local, but not international, journal.
- Higher incidence of diabetes in women might be due to history of gestational diabetes. This point should be more clearly investigated.
- Diabetes in this study included type 1, type 2 and other types of diabetes. Gestational diabetes even might be included. This point should be clarified.
- Figures 1 and 2 provide little information and should be removed.
- If the results were similar in the male and female subjects, Tables 3 and 4 can be presented as supplemental materials.
- Instead, supplementary Table 1 may be included in the main text. Also, a table comparing the parameters between subjects with and without diabetes should be provided.
- 1. answer rate should be described in the methods, but not the results section.
- In the abstract, AH should be spelled out at the first appearance.
- Same conclusion was repeated twice in the main text. This should be corrected.
Author Response
1-Comment: In this study, the authors examined the prevalence and risk factors for self-reported diabetes in the adult population of the Midwest region of Brazil. Using NHS data conducted in 2013, 7,519 adults were included in the study. The prevalence of diabetes was 6.5%. In the global sample, age, smoking, hypertension, dyslipidemia, overweight and obesity were independently associated with self-reported diabetes. Overall, the use of large cohort was a strength of the study, however, there are critiques as described below.
Answer: thanks for the careful evaluation of the manuscript. We sent the reformulated manuscript according to suggestions and answers to the questions below.
Comment: The findings of study were mostly confirmative and novelty of the study was not clear. The authors need to describe novelty of the study more precisely, or the manuscript may be suited to publish in more local, but not international, journal.
Answer: The NHS was a representative epidemiological survey for Brazil and its macro-regions developed between the years 2013 and 2014. The existing investigations on the survey report aggregated data for Brazil or information on the most developed macro-regions in Brazil. Thus, data on the epidemiology of DM in the Midwest region of the country had not yet been analyzed. This study shows results on the magnitude and factors associated with DM in the Midwestern macro-region, and can contribute to the implementation of effective preventive and control strategies for DM in this region to achieve the goals of the Global Plan for Coping with Chronic Diseases and the Health Plan. Strategic Actions to Combat Chronic Non-Communicable Diseases in Brazil, years 2011 to 2022.
We added this comment to the discussion.
3- Comment: Higher incidence of diabetes in women might be due to history of gestational diabetes. This point should be more clearly investigated.
Answer: Thanks for the observation. We agree with the reviewer, but as described in the method, we excluded women who reported gestational diabetes in their lifetime from the prevalence calculation.
4- Comment: Diabetes in this study included type 1, type 2 and other types of diabetes. Gestational diabetes even might be included. This point should be clarified.
Answer: As mentioned in the methodology section the NHS aimed to produce national data to characterize the health condition and lifestyle and access and use of healthcare services of the Brazilian population [16, 17]. Information regarding DM was obtained by asking the following question: “Were you diagnosed with diabetes by a doctor?” Women who reported gestational DM were excluded from the prevalence estimate.
Therefore, since this is a survey with a representative sample of the Brazilian population, to promote health policies in the area of chronic non-communicable diseases, the question was self-reported, and does not differentiate type 1 diabetes from type 2.
In addition, the participant was not asked about the family history of DM. Some studies were developed using the self-reported measure and it has good reliability to identify people with DM.
We agree with the reviewer that there are similar risk factors between the two types, but this is a limitation of our work and will be mentioned in the limitations section.
(References:
- Ning M, Zhang Q, Yang M. Comparison of self-reported and biomedical data on hypertension and diabetes: findings from the China Health and Retirement Longitudinal Study (CHARLS). BMJ Open [Internet]. 2016 Aug 11, 2020; 4; 6 (1): e009836. Avaiable from: https://pubmed.ncbi.nlm.nih.gov/26729390/.
- Peterson KL, Jacobs JP, Allender S, Alston LV, Nichols M. Characterizing the extent of misreporting of high blood pressure, high cholesterol, and diabetes using the Australian Health Survey. BMC Public Health [Internet]. 2016 Aug 11, 2020; 16: 695. Avaiable from: https://bmcpublichealth.biomedcentral.com/articles/10.1186/s12889-016-3389-y.
- Paalanen L, Koponen P, Laatikainen T, Tolonen H. Public health monitoring of hypertension, diabetes and elevated cholesterol: comparison of different data sources. Eur J Public Health [Internet]. 2018 Aug 11, 2020; 1; 28 (4): 754-765.Avaiable from: https://pubmed.ncbi.nlm.nih.gov/29462296/.
- Pastorino S, Richards M, Hardy Rebecca, Abington Jane, Wills Andrew et al. Validation of self-reported diagnosis of diabetes in the 1946 British birth cohort. 2015 Aug 11, 2020; 9 (5): 397–400. Avaiable from: https://www.ncbi.nlm.nih.gov/pmc/articles/PMC4582042/.
5-Comment: Figures 1 and 2 provide little information and should be removed.
Answer: thanks for the observation. We excluded Figure 2. However, we ask for permission to maintain Figure 1, which shows the location of the study's geographic area to locate the reader.
6-Comment: If the results were similar in the male and female subjects, Tables 3 and 4 can be presented as supplemental materials.
Answer: ok.
7-Comment: Instead, supplementary Table 1 may be included in the main text. Also, a table comparing the parameters between subjects with and without diabetes should be provided.
Answer: We believe that these are bivariate tables already in the text.
8-Comment: answer rate should be described in the methods, but not the results section.
Answer: thanks for observation. The response rate was transferred to the “materials and methods” section as noted.
9-Comment: In the abstract, AH should be spelled out at the first appearance.
Answer: thanks for observation. Due correction was carried out.
- Comment: Same conclusion was repeated twice in the main text. This should be corrected.
Answer: thanks for observation. Due correction was carried out.
Round 2
Reviewer 1 Report
Thanks for this very careful and thorough revision, addressing all points satisfactory.
I consider the paper now to be fit for publication.
Reviewer 2 Report
The authors addressed all my comments, then in my opinion the paper can be accepted in present form.
Reviewer 3 Report
The authors responded to the comments properly.